# Device-based measurement of physical activity and sedentary behaviour after critical illness: A scoping review

Darren Murphy[1]*, Annette Henderson[2], Judy M. Bradley[3], Bronwen Connolly[3], Jason J. Wilson[4‡], Brenda O'Neill[1‡]*

1 Institute of Nursing and Health Research, School of Health Sciences, Ulster University, Belfast, United Kingdom, 2 School of Health Science, Derry/ Londonderry campus, Ulster University, Derry/ Londonderry, United Kingdom, 3 Wellcome-Wolfson Institute for Experimental Medicine, School of Medicine, Dentistry and Biomedical Sciences, Queen's University, Belfast, United Kingdom, 4 School of Sport and Exercise Science, Derry/ Londonderry campus, Ulster University, Derry/ Londonderry, United Kingdom

‡ Joint senior authors
* murphy-d76@ulster.ac.uk (DM); b.oneill@ulster.ac.uk (BON)

## Abstract

### Introduction

Measuring and promoting physical activity could support rehabilitation and recovery after critical illness. In recent years, there has been an emergence of the use of devices to measure both physical activity and sedentary behaviour in this population. Understanding device selection and processes for data analysis could be helpful for future research and practice when used with this population.

### Aims

The aim of this review was to explore the current use of device-based physical activity instruments to measure physical activity and sedentary behaviour during and following critical illness.

### Methods

A scoping review was conducted which followed the Arksey and O'Malley (2005) framework. A comprehensive search of four electronic databases (Medline, ProQuest, Scopus and CINAHL) was conducted using pre-agreed search terms. Screening and data extraction was conducted by two independent reviewers. Data were analysed descriptively by summarising and describing results that linked to the review questions.

### Results

Twenty-two studies were included; the majority were observational (n=12), with one randomised control trial. Studies covered the continuum from intensive care

**Data availability statement:** All relevant data are within the manuscript and its Supporting Information files.

**Funding:** (DM) Funded by the Department for the Economy, Northern Ireland (DfE) PhD funding. The funders had no role in study design, data collection and analysis, decision to publish, or preparation of the manuscript.

**Competing interests:** The authors have declared that no competing interests exist.

admission to 18 months post-hospital discharge. A total of 11 devices were used to assess physical activity and sedentary behaviour, and many different processing decisions were used for data analysis. Physical activity levels were low in the intensive care unit and remained low following discharge from intensive care.

## Conclusion

The use of device-based measurement of physical activity and sedentary behaviour after critical illness is an emerging research area. While several devices are available, this review highlights the need for agreed and standardised protocol(s) to guide the processing and analysis of data. Investment is also needed to support the recovery of physical activity and the reduction of excessive sedentary behaviour following discharge from the hospital.

## Introduction

After critical illness, reduced function, mobility, and strength may negatively impact quality of life and capacity to perform routine daily tasks [1–3]. In addition, patients often have difficulty achieving pre-illness physical activity levels [4]. These functional limitations are likely to prevent survivors of critical illness from achieving the recommended weekly levels of physical activity. Adults are recommended to undertake ≥150 mins of moderate intensity, ≥75 minutes of vigorous-intensity physical activity, or a suitable combination of both [5]. Guidelines also recommend a reduction in sedentary behaviour [5]. The effects of high levels of sedentary behaviour have been well documented, with sedentary behaviour having negative effects on health and increasing the risk of mortality in older adults [6,7]. Negative physiological effects of sustained levels of sedentary behaviour include insulin resistance, vascular dysfunction and a reduction in muscle mass and strength [8]. Reallocation of the length of time spent in sedentary behaviour into moderate to vigorous physical activity (MVPA) can result in health benefits, with researchers reporting that changes of as little as 4–12 minutes can be useful [9]. The consequences of critical illness, as well as excessive sedentary behaviour, have the potential to impact patients' recovery and return to activity after hospital discharge. However, physical activity is not routinely measured in this population, and it is not clear which device/s should be used [10]. There is also limited data to quantify and describe levels of physical activity and sedentary behaviour after critical illness.

There is increasing awareness about the need to evaluate and promote rehabilitation and physical activity in the critical care population during and after hospitalisation [11]. A previous scoping review (n=7 studies) aimed to evaluate the use of wearable devices in the intensive care unit (ICU) and found that device-based physical activity measurement was used infrequently in people after critical illness [12]. However, this is an emerging area of research and several papers about physical activity after critical illness have been published since the Gluck *at al*. (2017) review [13,14]. Researchers and clinicians will need to be able to accurately assess physical activity

and sedentary behaviour levels to identify activity patterns across the trajectory of recovery, evaluate the prevalence of physical activity engagement, and assess the effectiveness of interventions designed to change physical activity and sedentary behaviour. It is anticipated that this review will advance knowledge about device-based instruments, and any challenges for the measurement of physical activity and sedentary behaviour in this population. This review will inform the potential use and implementation of device-based measurement of physical activity and sedentary behaviour following critical illness in future research and in clinical practice.

## Aims

This scoping review aimed to explore the current use of device-based physical activity instruments to measure physical activity and sedentary behaviour during and following critical illness. The objectives were, in patients following critical illness, to identify device-based instruments and the parameters used to describe patterns and levels of physical activity and sedentary behaviour, report device clinimetric properties; describe the relationship between physical activity and sedentary behaviour with other health-related outcomes and explore the levels and patterns of physical activity and sedentary behaviour across the trajectory of recovery.

## Methods

We chose a scoping review as the most appropriate review method in order to synthesise the evidence to address specific questions relating to the device-based measurement of physical activity and sedentary behaviour [15]. The scoping review applied the framework by Arksey and O'Malley [16], which was further refined by Levac [17]. The methodology is described in the study protocol (published in advance https://osf.io/278mc/) and is reported according to the Preferred Reporting Items for Systematic reviews and Meta-Analyses extension for Scoping Reviews (PRISMA-ScR) [18]. There was one deviation from the published protocol as two electronic databases, PubMed and Embase, were removed based on a recommendation from the subject librarian and were replaced with Scopus and ProQuest. This scoping review aimed to address the following questions:

1. What device-based instruments are currently used to measure physical activity and sedentary behaviour during and following critical illness?

2. What parameters have been used to describe patterns and levels of physical activity and sedentary behaviour in patients following critical illness?

3. What are the clinimetric properties (validity, reliability, responsiveness), user acceptability, and minimal clinically important differences (MCIDs) of device-based instruments to measure physical activity and sedentary behaviour specifically in patients following critical illness?

4. What is the relationship between physical activity and sedentary behaviour with other health-related outcomes (e.g., quality of life, hospital readmission, exercise capacity, function and emotional status) in patients following critical illness?

5. What are the levels and patterns of physical activity and sedentary behaviour in patients following critical illness across the trajectory of recovery?

## Search strategy

The research team collaborated with a research librarian to develop the search strategy, including the selection of the databases and refining the keywords and phrases. A comprehensive search of four electronic databases (Medline, ProQuest, Scopus and CINAHL) was conducted using the agreed-upon search terms. Medical subject headings (MeSh)

terms were expanded and included in the relevant databases (Supplement 1). The research team met to provide their experience and methodological expertise to further guide the identification of relevant studies.

Results were searched from inception to December 2023 and were limited to human research and English language.

## Eligibility criteria

Studies which reported the outcome of physical activity or sedentary behaviour in survivors of critical illness or ICU were included. The full inclusion criteria were as follows:

1. Adult patients admitted to an ICU (≥18 years old).

2. Patients who have been mechanically ventilated in ICU.

3. Critically ill patients previously admitted to an ICU, e.g., medical, surgical, neurological, cardiac and trauma diagnoses.

4. Physical activity monitoring that had commenced: during ICU stay, post-ICU discharge (on the ward) or post-hospital discharge.

5. Included some form of device-based physical activity measurement used to assess "physical activity" and/or "sedentary behaviour", including but not limited to accelerometers, pedometers, activPAL, ActiGraph, smartphone apps etc.

## Study selection

The search results from each of the databases were imported into Covidence, which was used as a screening and data extraction tool. Duplicates were removed automatically. Each study title and abstract from the systematic search was screened by two independent reviewers [DM and AH] against the inclusion and exclusion criteria. The full text was retrieved for all of those that appeared to meet the inclusion criteria, and for those where there was insufficient information regarding whether the criteria were met.

All full-text papers were then independently reviewed by the two reviewers [DM and AH] to determine inclusion. Any disagreements were addressed via discussions, or with the consultation of a third reviewer [BON or JW] when required.

## Data extraction and analysis

The studies were randomly divided between the two reviewers for completion of data extraction (DM & AH). The data were transferred to tables on an Excel database, which was drafted specifically to capture all the information required to answer the research questions. The data extracted included study characteristics, device details, parameters of physical activity and sedentary behaviour, correlations with other outcomes and levels of physical activity and sedentary behaviour. Once completed, the tables were reviewed by DM and AH to ensure all relevant data were captured and collated. Data were analysed descriptively by summarising and describing results that linked to the review questions. Thorough discussion took place among the review team to agree all results

## Results

### Study selection

The identification and selection of studies is summarised in Fig 1. There were 22 studies included.

### Study characteristics

Study characteristics are summarised in Table 1 (with further details in Supplemental S2 Table). The studies were undertaken in Europe (n=6), North America (n=8), South America (n=3) and Oceania (n=5). There was heterogeneity in the study designs utilised; the majority (n=12) were observational studies, with one randomised control trial [29], and one conference

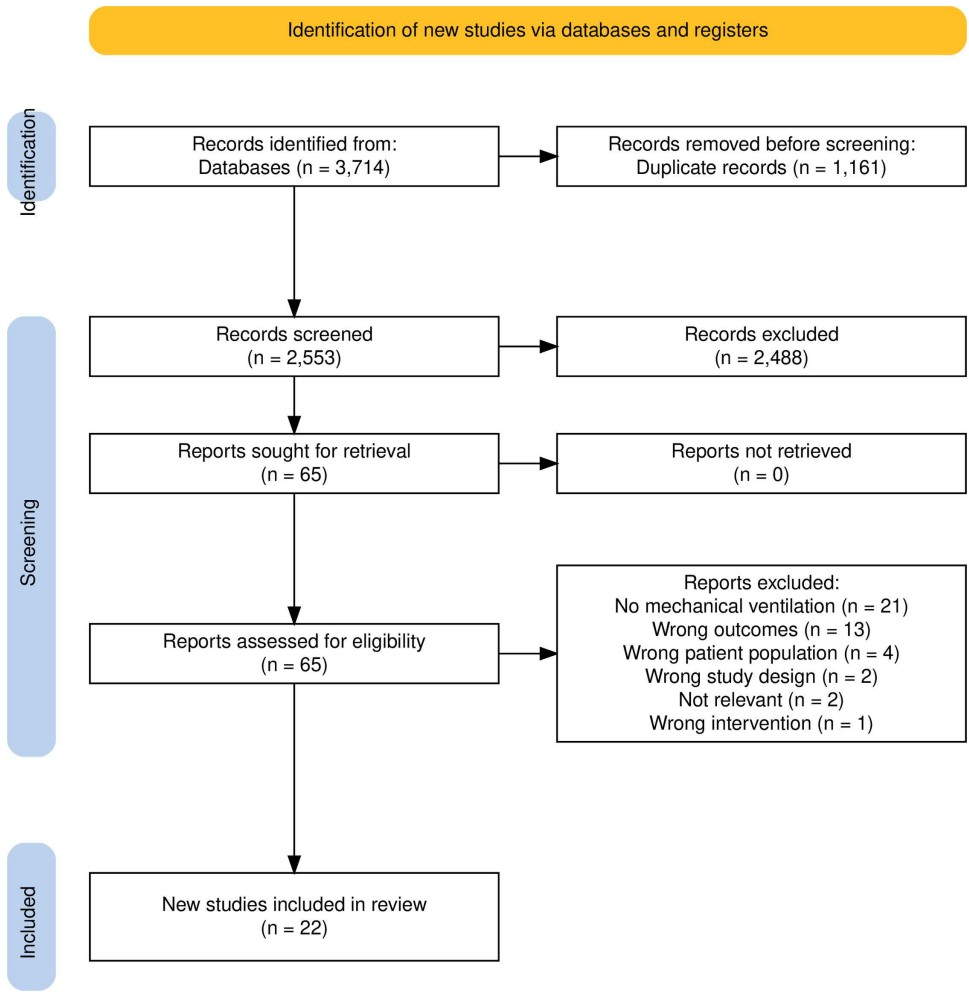

**Fig 1. Prisma flow diagram for selection of studies [19].**

abstract [22] included. Studies collected data across the continuum of recovery, with the earliest timepoint occurring at awakening in ICU [13,25] through to 18 months post-hospital discharge [24]. The years of publication were 2005–2022.

Several studies (n=14) provided data at one time point only, while six provided data across two time points and two studies provided data at three or more time points. Control or comparator groups were included in four studies [29,23,24,37].

## Population studied

Supplemental S2 Table summarises the characteristics of this heterogeneous population. Length of stay in the ICU varied from 4 [27] to 21 [28] days, and the duration of mechanical ventilation was reported in 17/22 (77%) studies. The population sample sizes in the included studies ranged between 8 [26] and 715 [37] participants.

## Device-based instruments used to measure physical activity and sedentary behaviour

Eleven different devices were used across the 22 studies (Table 1). Actiwatch Spectrum (n=5), activPAL3 (n=3) and the SenseWear armband (n=3) were the most commonly used devices. Location worn was dependent on the device used, and when reported, the most common was the wrist (n=9), thigh (n=4), and ankle (n=4). Ten studies reported device

**Table 1. Physical activity devices including details of wear time, activity parameters and data processing per device.**

| Author, year, aims, | Device, Location worn, Clinimetric properties. | Wear time. When to wear/not to wear | Activity Parameters | Device processing parameters |
|---|---|---|---|---|
| Grap et al., 2005 [20]. To determine the feasibility of continuous measurement of limb movement via wrist and ankle actigraphy in critically ill patients and to compare actigraphy measurements with observed activity, subjective scores on sedation-agitation scales and heart rate and blood pressure of patients. | Device: Mini Mitter, Actiwatch No of axis: Uni-axial Location worn: Non-dominant wrist and right ankle Clinimetric properties: N/A | Worn for the two-hour data collection period. | NR | Epoch length: 15 seconds Sampling frequency: NR |
| Winkelman, 2010 [21]. This study investigated whether selected cytokines vary between rest and activity periods in critically ill COPD patients. The secondary aims were to explore the types and duration of brief episodic activity. | Device: Mini Mitter, Actiwatch No of axis: Uni-axial Location worn: Dominant wrist Clinimetric properties: N/A | 24-48 hours, beginning with enrolment in the study. | Time (duration) of movement as well as intensity. | Epoch length: 1 minute Sampling frequency: NR |
| Beach et al., 2014 [22]. To describe PA levels during ICU admission and barriers to PA, to describe the relationship between PA, muscle strength and physical function at ICU discharge, to describe the relationship between muscle strength, physical function and ventilator-free days and between PA levels and ICU/hospital LOS. | NR | NR | NR | NR |
| Borges et al., 2015 [23]. The aim of the study was to quantify physical activity in daily life, muscle strength and exercise capacity in short and medium-term survivors of severe sepsis and shock. | Device: DynaPort MiniMod No of axis: Tri-axial Location worn: Lower back near L2 Clinimetric properties: N/A | Wear for 2 consecutive days. | Time spent, sitting, lying, standing, and walking, changes in body position and energy expenditure. | Epoch length: NR; Sampling frequency: NR |
| Mc Nelly et al., 2016 [24]. To explore the relationship between objective measures of PA, self-reported measures of physical HRQOL, and clinician-reported global functioning (frailty). To investigate the relationship between chronic disease status prior to critical illness and functional outcome | Device: SenseWear armband No of axis: Bi-axial Location worn: Upper arm, dominant/non-dominant not reported Clinimetric properties: Validity and acceptability | Wear daily for 5 days to include 1 weekend day, valid day ≥90% on body time per day for ≥ 5 days. | Number of steps, HRQoL, clinical frailty score. | Epoch length: NR; Sampling frequency: NR |
| Beach et al., 2017 [25]. Quantify levels of PA and evaluate the correlation of SWA-MF measurement of active and resting energy expenditure against the ICU mobility scale. Secondary outcomes, describe the feasibility of using the SWA-MF in the ICU setting, describe changes in physical function over the ICU stay and the relationship between PA duration and physical function. | Device: SenseWear armband No of axis: Bi-axial Location worn: Posterior aspect of the mid upper arm, dominant/non-dominant not reported Clinimetric properties: Acceptability reported | 24 hours per day for 5 days, valid day required minimum of 8 hours data for 3 consecutive days. | PA was defined as > 1 MET, number of steps, total energy expenditure, and active energy expenditure. | Epoch length: 1 minute; Sampling frequency: NR |
| Baldwin et al., 2018 [26]. To test the hypothesis that there would be an agreement between the Activ Pal accelerometer and direct observation in classifying sit-to-stand transitions, time spent standing, stepping, upright and sedentary | Device: ActivPAL3 No of axis: Tri-axial Location worn: Non-dominant thigh Clinimetric properties: Validity assessed | NR | Time spent, sitting, standing, or stepping. | Epoch length: NR Sampling frequency: NR |

*(Continued)*

**Table 1.** (Continued)

| Author, year, aims, | Device, Location worn, Clinimetric properties. | Wear time. When to wear/not to wear | Activity Parameters | Device processing parameters |
|---|---|---|---|---|
| Estrup et al., 2018 [27]. The primary outcome was the association between physical function 3 months after discharge from ICU measured by CPAx and activity levels during the first week at the ward measured by actigraphy. | Device: Micro Sleep watch No of axis: NR Location worn: Wrist, dominant/ non-dominant side not reported. Clinimetric properties: N/A | Worn for 7 days after ICU discharge. | Physical activity was analysed in several ways: Mean of activity counts per day; mean activity per hour during daytime (7 am-4 pm); mean difference between day and night; maximum activity in 1 hour on day 2; total activity count on day 2; and finally, daytime activity on day 2 | Epoch length: NR; Sampling frequency: NR |
| Anderson et al., 2019 [28]. Investigate the validity of the Actigraph GT3X accelerometer, quantify step count during self-selected distances and walking speeds in ward-based hospital patients recovering from critical illness | Device: ActiGraph GT3X No of axis: Tri-axial Location worn: One was attached to the anteromedial non-dominant thigh and the other to the lateral aspect of the non-dominant ankle (above lateral malleolus). Clinimetric properties: Validity assessed | For the walking aspect of the protocol. | Observed step count was compared to step count data captured by ankle and thigh-mounted accelerometer. | Epoch length: NR; Sampling frequency: NR |
| Schujmann et al., 2019 [29]. The aim was to investigate whether patients who participated in a mobility program in the ICU performed better on functional status, muscle, mobility, and respiratory assessments upon discharge than patients who received conventional physiotherapy. | Device: ActiGraph GT3X No of axis: Tri-axial Location worn: Dominant lower limb. Clinimetric properties: N/A | Time spent in ICU. The accelerometer was not removed until ICU discharge. | Amount of inactive time and total time in levels of light and moderate intensity of activity. | Epoch length: NR; Sampling frequency: NR |
| Baldwin et al., 2020 [13]. To examine the patterns of sedentary behaviour and physical activity in people during their hospital-based recovery from a critical illness. | Device: ActivPAL3 No of axis: Tri-axial) Location worn: Non-dominant thigh Device: GENEActiv No of axis: Tri-axial Location worn: Non-dominant wrist. Clinimetric properties: N/A | Wear length: 1 day Wear/ non-wear instructions: Worn continuously except during any off-unit/ward procedures or imaging. Invalid day if ≥6 h of non-wear. | ActivPAL 3: % time in sedentary behaviour; time spent upright (minutes); number of upright bouts ≥2 and ≥5 minutes; number of sit-to-stand transitions; number of sitting/ lying bouts ≥30 minutes and ≥2 hours) GENEActiv: % time spent inactive; time spent being active (minutes) | ActivPAL 3 Epoch length: 60 seconds; Sampling frequency: 20 Hz GENEActiv Epoch length: 60 seconds; Sampling frequency: 50 Hz |

*(Continued)*

**Table 1.** (Continued)

| Author, year, aims, | Device, Location worn, Clinimetric properties. | Wear time. When to wear/not to wear | Activity Parameters | Device processing parameters |
|---|---|---|---|---|
| Camus-Molina et al., 2020 [30]. Assess construct validity of the Chilean-Spanish version of FSS-ICU using 3 variables measured through continuous actigraphy (activity counts, activity time, inactivity time) in mechanically ventilated patients from their admission to ICU discharge. The secondary aim is to test the correlation of the FSS-ICU with MRC-SS ICU length of stay and duration. | Device: ActiGraph GT9X Link No of axis: Tri-axial Location worn: Right ankle Clinimetric properties: N/A | Daily for 24 hours from ICU admission to ICU discharge. The accelerometer was removed only for out-of-room clinical tests (i.e., MRI, CT, surgical procedures). Twice a day, patients' skin condition, battery level and functioning of accelerometers were checked. | Activity counts, activity time, and inactivity time were analysed. | Epoch length: 5 seconds; Sampling frequency: 90 Hz |
| Gupta et al., 2020 [31]. Use actigraphy to characterise inactivity and activity in critically ill patients | Device: Actiwatch Spectrum No of axis: Uni-axial Location worn: Right wrist, or if unavailable due to arterial line, then left wrist Clinimetric properties: N/A | 48 hours in ICU. | Steps not measured. Only epochs in terms of zero and non-zero activity epochs were recorded. | Epoch length: 30-second; Sampling frequency: NR |
| Elias et al., 2021 a [32]. Examine trends between post-ICU hourly activity counts and discharge disposition among hospitalised older ICU survivors. (Transitioning from ICU hospitalised older adults are inactive which may affect discharge outcomes) | Device: Actiwatch Spectrum No of axis: Uni-axial Location worn: Worn on the dominant wrist. Clinimetric properties: N/A | Early post-ICU hourly activity worn continuously 00:00 - 23:59. | Activity counts were generated for each 15-second epoch. | Epoch length: 15 seconds; Sampling frequency: NR |
| Elias et al., 2021 b [33]. Explore associations between post-ICU activity, sleep/rest and motor function among hospitalised older ICU survivors. (hospitalised older ICU survivors are often inactive and experience sleep disturbances). | Device: Actiwatch Spectrum No of axis: Uni-axial Location worn; Dominant wrist Clinimetric properties: N/A | Worn post-ICU (24–48hr) continuously for 2 night-time and 1 daytime data capture. | Activity counts were generated for each 15-second epoch. | Epoch length: 15 seconds; Sampling frequency: NR |
| Elias et al., 2021 c [34]. Explore the relationship between early post-ICU activity, discharge deposition and LOS. | Device: Actiwatch Spectrum No of axis: Uni-axial Location worn: Dominant wrist Clinimetric properties: N/A | Worn continuously throughout a 2 night/1 day observation period, beginning at the time of study enrolment. | Activity counts were generated for each 15-second epoch. | Epoch length: 15 seconds; Sampling frequency: NR |
| Gandotra et al., 2021 [14]. To quantify activity levels and functional status in critically ill patients as they transition from ICU to the wards and back to the community. | Device: Tractivity No of axis: Uni-axial Location worn: Ankle worn, dominant/non-dominant side not stated. Clinimetric properties: N/A | At time of enrolment to 1 week after discharge. (Not stated if 24hr wear or when removed). | Step counts at 4 measurement points, mean number steps: 3 days prior to ICU discharge, 3 days prior to hospital discharge, 1-3 days after hospital discharge, 4-6 days after hospital discharge. | Epoch length: NR; Sampling frequency: NR |

*(Continued)*

**Table 1.** (Continued)

| Author, year, aims, | Device, Location worn, Clinimetric properties. | Wear time. When to wear/not to wear | Activity Parameters | Device processing parameters |
|---|---|---|---|---|
| Munro et al., 2021 [35]. To explore nighttime sleep efficiency and total sleep time over 5 consecutive nighttime periods, the secondary objective was to report the daily activity ratio and the daily activity counts over 5 consecutive days. | Device: Actiwatch Spectrum No of axis: Uni-axial Location worn: Dominant wrist unless blocked by medical lines etc. Clinimetric properties: N/A | Wear for 5 consecutive days (120 hours). | Not specified, the study measured activity using a daily activity ratio (DAR) and hourly activity counts, definition of activity is not defined. | Epoch length: 15 seconds; Sampling frequency: NR |
| Lehmkuhl et al., 2022 [36]. This study aimed to describe the objectively assessed type, quantity and daily variation of physical activity among mechanically ventilated patients while in ICU. | Device: Axivity AX3 accelerometer No of axis: Tri-axial Location worn: Right thigh and upper body Clinimetric properties: Acceptability reported | Worn from inclusion to ICU discharge, Day considered invalid if <50% wear time per day. | Time spent lying, sitting, moving, in-bed cycling, standing, and walking. | Epoch length: NR; Sampling frequency: 25 Hz |
| Plekhanova et al., 2022 [37]. The study aims to describe accelerometer-assessed physical behaviours in patient's post-hospitalisation with COVID-19 | Device: GENEActiv No of axis: Tri-axial Location worn: Non-dominant wrist Clinimetric properties: N/A | Wear length, 14 days for 24 hours, valid day ≥ 16 hours per day. | Inactivity, LIPA, MVPA | Epoch length: 5 seconds; Sampling frequency: 30 Hz |
| Rollinson et al., 2022 [38]. This study aimed to measure the physical activity of patients with critical illness engaged in rehabilitation in the ICU and on the acute ward and report discharge destination, muscle strength and functional outcomes. | Device: SenseWear armband No of axis: Bi-axial Location worn: Left upper arm Clinimetric properties: N/A | Worn 7 days per week during daylight hours only from 08:00-18:00 until hospital discharge or after 14 days in the acute ward, whichever occurred first. The armband was not waterproof so was removed for showering activities. | PA levels >1.5 METs. | Epoch length: NR; Sampling frequency: NR |
| Van Bakel et al., 2022 [39]. The main aim of this study was to objectively assess PA, SB and sleep duration in patients with COVID-19 with moderate to severe illness that required hospitalisation. The measurements took place at 3 and 6 months after discharge. | Device: ActivPAL3 No of axis: Tri-axial Location worn: Thigh worn, dominant/non-dominant not stated. Clinimetric properties: N/A | NR | LIPA, MVPA or SB: seated, reclined or lying. | Epoch length: NR; Sampling frequency: 20 Hz |

**Abbreviations:** CPAx: Chelsea Critical Care Physical Assessment Tool, COPD: Chronic Obstructive Pulmonary Disease, FSS-ICU: Functional Status Score for the Intensive Care Unit, HRQOL: Health-related quality of life, ICU: Intensive Care Unit, LIPA: Light intensity physical activity, MRC-SS: Medical Research Council Sum Score, MVPA: Moderate to vigorous physical activity, N/A: Not applicable, NR: Not recorded, PA: Physical activity, SB: Sedentary behaviour

placement: this was on the dominant (n=6) or non-dominant limb (n=5). Wear-time duration was reported in 18/22 studies (81%). Wear-time duration varied depending on the aims of the study. When validation studies were excluded, wear-time ranged from two hours [20] to 14 days [37]. In studies where accelerometry was used for more than one day, for analysis the parameters for a valid day varied from a minimum of 8 hours of data per day [25] to ≥90% of on-body time per day for ≥5 days [24].

The sampling frequencies used varied across studies, with many not reporting this (n=16), while other studies used 20–30 Hz (n=4), 50 Hz (n=1) and 90 Hz (n=1). Epoch length also varied, with the majority of studies not reporting this (n=12), while 60-second (n=4), 5-second (n=2), 15-second (n=2) and 30-second (n=1) epochs were reported in the other studies (Table 1).

### Parameters to describe patterns and levels of physical activity and sedentary behaviour

Various parameters (n=4) have been used to describe the levels of physical activity and sedentary behaviour, including step counts, activity counts, time in activity (total daily time, percentage of time, and number of bouts), and intensity of physical activity (Table 1).

### Clinimetric properties of the device-based instruments

Clinimetric properties were measured in five studies [25,24,26,28,36] (Table 1). Validity was assessed in three studies [24,26,28]. The ActivPAL demonstrated good validity when recording sit-to-stand transitions, time spent standing and sedentary behaviour when compared to direct observation, however, the validity of the ActivPAL to record step counts could not be confirmed [26]. In the hospital ward following ICU discharge, an ankle-placed ActiGraph was valid in measuring step counts, with the intraclass correlation (ICC) equalling 0.99, while step counts were shown to be underestimated using a thigh-worn device [28]. Accelerometry-measured step counts were shown to be a valid method of assessing frailty and health-related quality of life (HRQOL), with no floor or ceiling effect using the Sense-Wear device [24].

Reliability and responsiveness were not reported in any of the studies. Also, no studies reported on the minimal clinically important difference (MCID) for any metric with physical activity or sedentary behaviour. User acceptability was reported in three studies [25,24,36]. The SenseWear device was well-tolerated by participants [24] with 97% of participants wearing it for five days in the ICU [25]. The Axivity device was pilot tested on five participants, with no adverse effects noted while wearing it [36].

### Association of physical activity and sedentary behaviour with other health-related parameters

Physical activity data were compared to other health-related outcomes in 15/22 studies (68%) (S3 Table).

**Physical function.** Eight studies [13,14,22,24,25,29,27,33] showed positive correlations between physical activity levels and physical function. However, Rollinson *et al.* [38] did not show any significant associations. In addition, Baldwin *et al.* [13] showed there was no association between sedentary behaviour and physical function [13].

**Exercise capacity.** A strong correlation was reported between physical activity and exercise capacity in the ICU and in the ward (r=0.728; p<0.001) [38].

**Muscle strength.** Less time sitting or lying, more minutes spent upright and an increased number of STS transitions were associated with better muscle strength (MRC-ss and handgrip) at ICU discharge, however, there were no associations found at hospital discharge [13]. There was a positive association between post-ICU daytime activity and grip strength (r²=0.689, p<0.001) [33].

**HRQOL.** Daily step count was shown to correlate with the physical function component of the SF-36 (r²=0.51; p<0.01) and a weaker correlation (r²=0.25; p<0.01) was shown with the physical component score of the SF-36 [24].

**Hospital LOS.** Two studies [13] and [22] showed no associations between physical activity and/or sedentary behaviour and hospital length of stay, however, Elias and colleagues showed there was a significant relationship between post-ICU daytime activity and LOS [34].

**Other.** Moderate correlations were shown between the Riker Sedation and Agitation scale (SAS) (rho=0.601; p<0.005) and the Richmond Agitation Sedation Scale (RASS) r=0.58 & r=0.52) and physical activity levels [22,20].

Lower levels of symptom severity, excluding cognition and anxiety, were associated with physical activity (p<0.05), with those experiencing the most severe acute illness having 1–2 mg lower volume of physical activity (p=0.045) and less time spent in MVPA (p=0.032) [37].

No studies explored the association between physical activity and/or sedentary behaviour with hospital readmissions.

## Levels and patterns of physical activity and sedentary behaviour

Physical activity and sedentary behaviour patterns were reported from awakening in ICU [25,26] up to 18 months post-discharge [24] in Table 2.

**Step counts.** Step counts were reported in 8/22 studies [14,22–25,29,26,28] Step counts were initially low but increased progressively from ICU to hospital discharge and home recovery. After awakening in ICU, ambulation was almost non-existent, with a median of four steps/day [25]. Daily step counts gradually increased to 95 at three days before ICU discharge, and 257 three days before hospital discharge [14]. There were minimal increases in daily step counts from 1,223 steps at three days and 1,278 steps up to eight days after hospital discharge [14]. At 18 months post-hospital discharge, daily step counts had risen to 5,803 steps, which was still significantly lower (p<0.001) than healthy age-matched controls (i.e., 11,735 steps) [24].

**Activity counts.** Activity counts were reported in nine studies [20,21,27,31,33,30,32,34,35]. At awakening, a mean of 31,590 counts per day was reported, increasing to a mean of 50,483 counts per day at ICU discharge [30]. Activity counts averaged 2,233 per hour, within 24–48 hours of ICU discharge, where 1,140 counts per day equate to resting or lying down while awake [32]. Following discharge from the ICU, mean activity counts during the daytime (07:00-16:00) were shown to be 43,699 [27].

Eight studies reported on the time spent being physically active [13,22,25,37,36,38,30,39]. During the ICU admission, time spent being physically active was low, with values of 17.8, 16.8 & 9.3 minutes per day [22,25,30]. This time increased at ICU discharge to 35.5 minutes per day [30], and to the hospital ward was 52.8 minutes per day [38]. A similar trend was shown by Baldwin [13] and colleagues, where time spent being physically active (>200g/min) increased from awakening (27 minutes) to 61 minutes at ICU discharge and to 96.8 minutes at hospital discharge [13].

**Sedentary behaviour.** Sedentary behaviour was reported in eight studies [13,29,23,37,31,36,30,39]. These studies show the trajectory and change in sedentary behaviour over time, from awakening in the ICU [13,30] to eight months post-hospital discharge [37]. High levels of sedentary behaviour and inactivity were reported in the ICU. At awakening, Baldwin et al. [13] reported that their participants spent 98.1% of their time sedentary.

At ICU discharge, many participants typically spent 97.5% of their day being inactive [30]. Those in the study by Baldwin et al. [13] reduced their percentage of time spent inactive to 95.7% at ICU discharge, while this number reduced further to 93% at hospital discharge. At three months post-hospital discharge, sitting or lying time was reduced to 58% (from 89.2% in the hospital ward) by those observed by Borges et al. [23].

## Discussion

This scoping review has demonstrated that a wide range of devices have been used to measure physical activity and sedentary behaviour. There is a clear lack of standardisation regarding the data processing methods and there are no consistent parameters used when reporting physical activity levels or sedentary behaviour. There are some associations between physical activity and other outcomes such as physical function, strength, and exercise capacity. Overall, in people who have been admitted to ICU for critical illness, the levels of physical activity are generally low in ICU, on hospital wards and in the 18-month period after hospital discharge.

**Table 2. Measured Physical Activity and Sedentary Behaviour.**

| Study | Location | Step counts, Activity counts | Time in activity | Sedentary behaviour |
|---|---|---|---|---|
| Grap *et al.*, 2005 [20]. | ICU | **Total activity count for 15 min epoch**: **Wrist:** Mean 418±591.7 counts, range 0–2903, median 157.5, IQR 11.3,571.3 counts. **Ankle:** Mean 147.0±387.3 counts, range 0–3593; median 38, IQR 3.0-132.3 counts | N/A | N/A |
| Winkelman, 2010 [21]. | ICU and step-down unit | **Activity counts:** Day 1: 98 counts; Day 2: 124 counts | N/A | N/A |
| Beach *et al.*, 2014 [22]. | ICU | **Daily step count (steps/day):** Median 3.5 [IQR 1.4–5.9] steps/day | **Duration of PA (minutes/day):** Median 9.3 [IQR 0.4–151.2] minutes/day, METs: 0.9 [IQR 0.8–1.0] minutes/day | N/A |
| Borges *et al.*, 2015 [23]. | ICU to 3 months post-hospital discharge. | N/A | Percentage walking time was reduced in septic patients compared to the healthy control at ICU (1.9±1.6% vs 10.1±4.4%, p < 0.05) and at 3 months (6.3±3.0% vs 10.1±4.4%, p < 0.05) [Each per cent represents 7.2 minutes/day]; Sepsis patients had a lower walking intensity compared to control after hospital discharge, (2.1 ± 0.3 vs 2.5 ± 0.4 m/s$^2$, p <0.05). At 3 months, 65% of septic patients walked 30 minutes per day compared to 85% in healthy subjects. | Participants spent 89.2% of their ICU time sitting or lying down. After hospital discharge, this was reduced to 58% |
| McNelly *et al.*, 2016 [24]. | 18 months post-hospital discharge. | **Daily step count (steps/day)**: 5,803 (95%CI 4792–6813) was lower than in controls 11735 (95%CI 10928–12542) p<0.001 and lower in those with chronic pre-existing chronic disease 2989 95%CI 776–5201 than those without 7737 95%CI 4907–10567 p=0.013. | N/A | N/A |
| Beach *et al.*, 2017 [25]. | ICU | **Daily step count (steps/day):** Median: 4 [1.2–7] | **PA time (minutes per day):** Median of 16.8 [0.6–152.4] minutes/day **Participants undertaking PA >100 minutes of activity per day:** 1 (2%) participants on day 1 and 14 (25%) of participants on day 3. | N/A |
| Estrup *et al.*, 2018 [27]. | Ward following discharge from the ICU | **The median of the mean daily activity counts:** 104,634 (IQR 65,569–165,624) counts/day **Mean activity during daytime:** 43,699 counts/day (Daytime hours: 07:00-16:00) | N/A | N/A |
| Schujmann *et al.*, 2019 [29]. | N/A | **Daily step counts (steps/day):** At ICU discharge: Control: 591 ± 403 steps/day; Intervention: 1539 ± 966 steps/day | **Percentage of time in PA:** Light - Control: 3.9±1.9%, Intervention: 6.4±2.4%; Moderate - Control: 0.3±0.2%, Intervention: 1.01±0.60%; Intense - Control: 0.03±0.02%, Intervention: 0.15±0.10%. | **Percentage of time inactive:** Control: 95.7±2.0%; Intervention: 92.3±2.8% |

*(Continued)*

 

| Study | Location | Step counts, Activity counts | Time in activity | Sedentary behaviour |
|---|---|---|---|---|
| Baldwin *et al.*, 2020- [13]. | ICU, followed by hospital ward to discharge from hospital | N/A | **Number of activity bouts:**<br>At awakening:<br>≥2 minutes 2.9±5.4 bouts;<br>≥5 minutes 1.1±2.8 bouts;<br>At ICU discharge:<br>≥2 minutes 9.2±12.7 bouts;<br>≥ 5 minutes 3.2±4.8 bouts<br>At hospital discharge:<br>≥2 minutes 17.7±14.8 bouts<br>≥5 minutes 5.8±5.9 bouts<br>**Time spent in activity (mins) (≥200g/minute):**<br>At awakening: 27.0±54.7 minutes<br>At ICU discharge: 61.1±98.8 minutes<br>At Hospital discharge: 96.8±85.1 minutes | **Percentage of time in SB:**<br>At awakening: 98.1%<br>At ICU discharge: 95.7%<br>At hospital discharge: 93%<br>**No. of lying/sitting bouts:**<br>At awakening:<br>≥30 minutes: 2.5±2.1 bouts;<br>≥2 hours: 1.9±1.4 bouts;<br>At ICU discharge:<br>≥30 minutes: 5.1±3.3 bouts,<br>≥2 hours: 3.0±1.5 bouts;<br>At hospital discharge:<br>≥30 minutes: 9.3±4.5 bouts<br>≥2 hours: 3.4±1.5 bouts<br>**Number of upright bouts:**<br>At awakening:<br>≥2 minutes 0.9±2.4 bouts;<br>≥5 minutes 0.3±1.0 bouts;<br>At ICU discharge:<br>≥2 minutes 2.3±3.3 bouts;<br>≥5 minutes 0.6±1.1 bouts;<br>At hospital discharge:<br>≥2 minutes 7.2±4.7 bouts;<br>≥5 minutes 2.9±2.7 bouts |
| Camus-Molina *et al.*, 2020 [30]. | ICU | **Activity counts:**<br>Awakening: 99,396 (31,707–193,692) counts ICU discharge: 309,104 (133,737–557,149) counts<br>**Activity counts per day:**<br>Awakening: 31,590 (21,275–52,219) counts/day<br>ICU discharge: 50,483 (38,073–88,649) counts/day | **Activity (minutes/day):**<br>Awakening: 17.8 (11.5–26.9) minutes/day<br>ICU discharge: 35.5 (25.5–57.1) minutes/day | **Inactivity time (minutes/day):** Awakening: 1422 (1413–1429) minutes/day;<br>ICU discharge: 1405 (1382.9–1414.5) minutes/day;<br>Total actigraphy recording time (median) from admission to ICU discharge: 5.48 days (3.33–8.56);<br>0.15 days (0.06–0.25) equated to activity while 5.33 days (3.26–8.37) were inactive.<br>2.5% of time active, 97.5% of time inactive. |
| Elias *et al.*, 2021 [32]. | Ward within 24 hours of ICU discharge | Post-ICU hourly activity averaged 2,233±569 counts per hour. | N/A | N/A |
| Elias *et al.*, 2021b [33]. | Ward within 24 hours of ICU discharge | The mean post-ICU 24-hour activity was 39.7±27.2 counts/minute. | N/A | N/A |
| Elias *et al.*, 2021c [34]. | Ward within 24 hours of ICU discharge | Post-ICU daytime activity averaged 41.21 activity counts/minute (SD 28.24, range 5.8–91.2 [median=31.25, IQR 48.37] and night-time activity averaged 29.27 activity counts/minute (SD 28.98, range 4 to 137.52, median 17.76; IQR 36.35) | N/A | N/A |
| Gandotra *et al.*, 2021 [14]. | ICU, hospital ward, 7 days post-hospital discharge | **Daily step count (steps/day):**<br>Prior to ICU discharge: 95 steps/day [95% CI=15–173],<br>Prior to hospital discharge: 257 steps/day [95% CI= 114–400];<br>After hospital discharge: 1223 steps/day [95% CI= 376–2070]<br>4-6 days after discharge: 1278 steps/day [95% CI = 349–2207] | N/A | N/A |

*(Continued)*

| Study | Location | Step counts, Activity counts | Time in activity | Sedentary behaviour |
|-------|----------|------------------------------|------------------|---------------------|
| Munro *et al.*, 2021 [35]. | ICU | Most hourly activity <1,000 counts | The mean daytime activity ratio over the 5 days was 66.5±19.2%; A daily activity ratio occurrence >80% was 17%. | N/A |
| Lehmkuhl *et al.*, 2022 [36]. | ICU | N/A | **Moving time (minutes/day):** Median (IQR): 7 (3–11) minutes/day, 0–63 range; **Bed cycling time (minutes/day):** Median (IQR): 3 (0–20) minutes/day, 0–125 range. **Standing time (minutes/day):** Median (IQR):1 (0–3) minutes/day, 0–71 range; **Walking time (minutes/day):** Median (IQR): 0 (0–0.02) minutes/day, 0–0.25 range. 60 minutes/day standing, moving, walking, or bicycling while in ICU. | **Lying time (minutes/day):** Median, IQR: 1206 (917–1336); 199–1432 range; **Sitting time (minutes/day):** Median, IQR: 179 (52–381) minutes/day, 0–1172 minutes/day range. MV patients spent 1200 minutes/day lying in bed, 180 minutes sitting while in ICU. |
| Plekhanova *et al.*, 2022 [37]. | 8 months post-hospital discharge | N/A | **Time spent in LIPA (minutes/day):** Women: 149.8±54.9 minutes/day Men:138.8±51.4 minutes/day **Time spent in MVPA (minutes/day):** Women:14.9±14.8 minutes/day Men: 21.1±22.3 minutes/day **Intensity of the most active (mg):** Women: 10 minutes, 61.0 ± 39.0 mg; 30 minutes, 40.4 ± 25.4 mg Men: 10 minutes, 73.6 ± 64.0 mg, 30 minutes, 49.6 ± 47.8 mg **Percentage of participants meeting >150 minutes of MVPA per week:** Women: 68 (26.9%); Men: 172 (37.2%) **Number of days with a 10-minute moderate-intensity bout of PA:** 0 days: Women: 139 (55.8%); Men: 199 (44.2%) 1 day: Women: 48 (19.3%); Men: 82 (18.2%) 2 days: Women: 28 (11.2%); Men: 58 (12.9%) 3+ days: Women: 34 (13.7%); Men: 111 (24.7%) **Number of days with a 30-minute moderate-intensity bout of PA:** 0 days: Women: 191 (76.7%); Men: 285 (63.3%) 1 day: Women; 33 (13.3%); Men; 68 (15.1%) 2 days: Women: 14 (5.6%); Men: 45 (10%) 3+ days: Women: 11 (4.4%); Men: 52 (11.6%) | **Inactive time (hours/day):** Women: 12.1±1.7 hours/day Men: 12.6±1.7 hours/day **Sleep time (hours/day):** Women: 7.2±1.1 hours/day Men: 6.9±1.1 hours/day |
| Rollinson *et al.*, 2022 [38]. | ICU to hospital ward | N/A | **Time spent in PA >1.5METs (minutes/day):** ICU: 17.8(22.8) minutes/day; Ward: 52.8(51.2) minutes/day, Mean difference (95% CI): 35 (23.8–46.1) p<0.001 | N/A |

*(Continued)*

**Table 2.** (Continued)

| Study | Location | Step counts, Activity counts | Time in activity | Sedentary behaviour |
|---|---|---|---|---|
| Van Bakel *et al.,* 2022 [39]. | 3-6 months post-hospital discharge | N/A | **Activity time (hours/day):**<br>Total cohort (n=37):<br>MVPA (Median, IQR): 1.0, 0.8-11.4 hours/day;<br>LIPA (Median, IQR): 4.2, 3.2-5.3 hours/day<br>ICU population N=13:<br>LIPA (Mean±SD): 4.2±1.4 hours/day;<br>MVPA (Median, IQR): 1.2, 0.7-1.3 hours/day | **Sitting time (hours/day):**<br>Total cohort (N=37):<br>Median, IQR: 9.8, 8.7-11.2 hours/day;<br>ICU population N=13:<br>Mean±SD 9.9 ± 1.6 hours/day |

**Abbreviations:** 95% CI: 95% confidence interval, ICU: Intensive Care Unit, IQR: Interquartile range, LIPA: Light intensity physical activity; MVPA: Moderate to vigorous physical activity, N/A: Not applicable, PA; Physical activity, SB: Sedentary behaviour

## Device-based instruments used to measure physical activity and sedentary behaviour

The most common devices used to measure physical activity and sedentary behaviour included the Actiwatch Spectrum, activPAL and SenseWear Armband. This finding reflects the wider literature related to accelerometry measurement in various clinical populations. However, researchers and clinicians should also be aware of the strengths and weaknesses of each of the devices. For example, a thigh-worn activPAL device may be more suited to tracking and recording sedentary behaviour and transitions from various positions, while potentially underestimating step counts [13]. If the goal is to measure positional changes, choosing this device or similar would be more appropriate. Future literature to support the use of a particular device and the protocols for wearing the device, including for example, location worn, should be based on research in the critical care population as opposed to healthy populations or other cohorts to be most applicable [40,41]. While various accelerometers have been utilised in individuals admitted to ICU, it is important for researchers and clinicians to be aware that there is likely to be a certain amount of measurement error with each device. For example, Baldwin and colleagues [26] compared activPAL-measured outcome variables with direct observation in ICU survivors. The authors found that although sit-to-stand transitions, sedentary behaviour and upright time were well-recognised by the activPAL, there were under-estimations of standing and stepping time compared to direct observation (22% and 19%, respectively). In healthy populations, free-living step counts have been shown to vary across seven activity monitors by around 1,700 steps/day (ranging from 1265 to 2275 steps/day) [42]. This shows that even though two activity monitors are supposedly measuring the same variable, there are likely to be error margins to consider. Unfortunately, few studies have currently explored validity and reliability of different devices in ICU survivors.

## Parameters of physical activity and sedentary behaviour

There are inconsistencies across the studies when defining physical activity and sedentary behaviour, the intensity of activity, and the thresholds used for the analysis of data on sedentary behaviour, making comparisons between studies difficult [43]. For the measurement of physical activity and sedentary behaviour, specific activity counts, milligravity (mg) units and METs have all been used. For METs, previous research has determined a cut-off point of <1.5 METs to define levels of sedentary behaviour, while 3 METs is commonly used as the cut-off point of LIPA/MVPA [44–46]. Survivors of ICU are often left with reduced exercise capacity and reduced levels of oxygen consumption, possibly due to mitochondrial dysfunction [47]. Therefore, it may be more appropriate to use lower MET values, such as those used in older populations, which could be as low as 2.8 ml/O2/kg [48–50] to avoid underestimation of the intensity of effort [51]. This was considered in one study [25], where a value of 1 MET was used (as opposed to 1.5 METs) as the threshold for physical activity. There are also other processing decisions that must be considered, such as epoch length, device placement and sampling frequency.

An epoch is defined as the time-period which allows data to be grouped together for analysis and usually varies from 1–60 seconds. Using different epoch lengths can affect the results by either over- or under-estimating physical activity and sedentary behaviour [52,53]. For example, the use of a 10-second epoch has been shown to increase the total daily sedentary behaviour time of older adults by almost 80 minutes when compared to a 60-second epoch [53]. The optimal epoch length for data analysis for assessing device-based physical activity within the critical care population has not been determined. In addition, the location the device is worn (e.g., thigh, wrist) alongside the dominant/non-dominant side worn have been shown to impact on important physical activity parameters, such as step counts. For example, daily steps have been shown to be significantly different between wrist and hip locations in adults wearing a pedometer [54], while another study [55] demonstrated dominant wrist placement resulted in 1,253 more daily steps compared to the non-dominant side (p=0.006).

The use of various sampling frequencies can also have an impact on the results and classification of physical activity and sedentary behaviour [56–58]. When using an ActiGraph GT3X, it has been recommended to use multiples of 30Hz [56,58], however, it is worth noting that using multiples of 30 may lead to filtering out of signals generated by vigorous activity and may result in an underestimation of vigorous activity [56].

The current heterogeneous approach to selecting and reporting parameters for the measurement of physical activity and sedentary behaviour is a clear limitation of the existing body of research, as this impacts on the ability to compare and contrast study results or undertake meta-analysis. There is a need to clearly define and establish cut-points across the different MET thresholds, epoch lengths and sampling frequencies for analysis of physical activity data that relate to ICU survivors, as this would lead to a more consistent approach to measuring and reporting physical activity and sedentary behaviour. Table 3 provides a summary of some considerations and suggestions to help guide the use of device-based measurement in future critical care research and practice.

To overcome the challenges of measuring step counts in those with slow gait speed, and misclassification with standing, a low-frequency filter may be used (LFE). However, it is important to note that the use of a LFE may end up being overly sensitive. Some studies have demonstrated inflated step counts in adults when using the LFE [53,66,67], in some instances by up to 5,502 steps per day [53,66]. It is worth noting that Anderson *et al.* [28] used a LFE filter and concluded it provided a valid measurement of step counts from an ankle-worn ActiGraph in hospital inpatients recovering from critical illness. In line with the recommendations for the COPD population, validation of devices to accurately measure step count using the LFE should be determined before use in the critical illness population [67].

Decisions on minimum daily wear time also need to be carefully considered, to appropriately represent activity and sedentary data while also maintaining sample sizes [53,58]. A valid daily wear time of ≥600 minutes (Table 3) has been suggested [53], and the results of this review show that only four studies met this criterion [13,24,37,36].

### Clinimetric properties

An understanding of the clinimetric properties of devices used in the ICU population is important to help guide the selection of the appropriate device for research. While studies have shown promise in validating devices for the critical care population when used under controlled conditions [13,28], researchers also cite validation studies from other populations (COPD, older adults) and this might not be the best approach. No studies in this review reported an MCID for physical activity or sedentary behaviour variables in this population, making it difficult to establish the change that is important to the patient or to establish the clinical effectiveness of an intervention [68,69]. This gap needs to be addressed in future research.

### Relationship between physical activity and sedentary behaviour with other health-related outcomes

In this review, levels of physical activity were shown to correlate with physical function, which provides evidence of the validity of physical activity as a construct [13,14,22,24,25,29,27,33]. However, correlations between physical activity and other outcome measures such as cognitive function and frailty, that are recommended in core outcome sets, have not

**Table 3. Practical considerations and suggestions to guide device based physical activity measurement and analysis for the critical care population.**

| Question or consideration and suggested guidance |
| --- |
| **Which device should I use?** |
| The most validated devices commonly used across various populations are ActiGraph [29,28,30], ActivPal [13,26,39] and Axivity [36]. |
| **Where should I place the device?** |
| For optimal compliance: Wrist [58]<br>Optimal position for sedentary behaviour: thigh [58] |
| **Should I place on the dominant or non-dominant side?** |
| Depends on the device used. E.g. ActivPal non-dominant side [13,26,39] Main issue is to ensure consistent location is maintained. |
| **What should I provide to the patient/participant?** |
| Introduce the device to the participant, explaining what it does and does not do.<br>Provide a brief instruction manual, including information such as proper fitting of the device, whether to remove it for sleep, and information on items such as the meaning of different light alerts.<br>Recommend the participant fill in an activity monitor diary, to aid memory about when the device is worn and to record any error warning. This will also help understand when the device has been removed for data processing.<br>If meeting the participant after they have worn the device in free-living conditions, ask them to not take off the device, but wear it into the clinic/study visit location. |
| **How should I introduce the device to the participant to ensure optimal data collection?** |
| Follow the device manufacturer's guidelines for fitting the device. Also, refer to recommendations developed for other relevant clinical populations [59, 60].<br>Researchers should publish the procedure/protocol they used with the device in their study so that it is visible and will allow for replication of the study [61]. |
| **How many days should the device be worn for?** |
| Optimal recommendation: 7 days [58]<br>Minimum for a valid data set:<br>-Post ICU: ≥2 days [23,31].<br>-Free-living: ≥4 days (1 Weekend) [58] |
| **What is considered a valid day?** |
| Regarding 24-hour protocols, there is currently no consensus, however, minimum values have included ≥10 hours [62,63] and ≥20 hours [9,64]<br>If using a waking hours-only protocol, the optimal/best practice should be ≥10 hours [53,59]. |
| **What algorithms should be used to calculate non-wear time?** |
| This will depend on the selected device, the manufacturer's instructions, and the age of the cohort [58,65]. |
| **What outcomes should I consider?** |
| Ideally, 24-hour health behaviours should be considered (i.e., physical activity, sleep, and sedentary behaviour). |
| **How should sedentary behaviour be expressed?** |
| The daily duration (time) and expressed as a percentage of wear time alongside daily time spent in prolonged bouts (e.g., 30 minutes) [13,23,39]. |
| **How should physical activity be expressed?** |
| Overall summaries (e,g, step counts, vector magnitude units), daily durations, expressed as a percentage of wear-time, in light and moderate-vigorous physical activity, as well as considering the number and time spent in different bout lengths (e.g., 10 minutes) [13,29]. |
| **What thresholds should I use for determining sedentary behaviour and intensities in different physical activities?** |
| This depends on several factors, such as the age profile of the population as well as the device used and the device placement [51,58,65]. |

been extensively studied [70]. It is worth noting that not all studies showed significant correlations of physical activity with all the same health-related outcome measures. Potential reasons for this could be down to the specific device which was used in each study, but also the test used to collect the health-related outcome measure. For example, physical function was measured using a battery of four tests by Rollinson *et al.* [38], while Estrup *et al.* [27] used the Chelsea Critical Care Physical Assessment tool (CPAx) which assesses ten aspects of physical function. These two studies had contrasting findings when it came to the association of physical function with physical activity.

## Levels of physical activity and sedentary behaviour

Physical activity levels in ICU survivors are low compared to healthy populations, even up to 18 months post-discharge [24], with a high prevalence of sedentary behaviour. This could contribute to the negative long-term effects of sedentary behaviour that have been well-documented [8]. While it is important to increase levels of physical activity in ICU survivors [11], efforts should also be made to encourage less sedentary behaviour where possible [71,72]. Prolonged periods of sedentary behaviour when recumbent (lying) may result in little-to-no activity counts being recorded on the device and may give the impression of sleep. Similar challenges were found in the COPD population, and the algorithm developed for using the ActiGraph wGT3X-BT accelerometer to help determine when the person may be sleeping or indeed waking up could be useful [62].

Only one of the 22 included studies was a randomised control trial investigating the effect of an intervention (a mobility programme) on physical activity in a cohort of patients within 48 hours of admission to ICU [18]. Improvements reported in the study were consistent with findings supporting the benefits of early mobilisation in the ICU [73], therefore, accelerometry could be an applicable outcome in future research exploring effects of early mobilisation in the ICU or investigating the effectiveness of rehabilitation interventions after ICU discharge. High-quality experimental studies are required to establish cause and effect relationships between physical activity and sedentary behaviour with important patient outcomes. At present, nearly all research is observational in nature. Therefore, there is a clear need for future experimental studies exploring the impact of changing physical activity and sedentary behaviour on important patient outcomes. Investigation is also needed to test the best way to support recovery of physical activity and reduction of excessive sedentary behaviour following discharge from hospital.

The strengths of this scoping review include a comprehensive search strategy and pre-identified rigorous methodology (15,16) as well as the inclusion of studies across the entire recovery trajectory following ICU. The review protocol was registered *a priori* to ensure transparency. Limitations of this scoping review were that we did not contact authors to ask whether further details about their data processing could be made available and our results were limited by the lack of a standardised approach to measurement used in the included papers making it difficult to compare across studies.

In future, researchers and clinicians will need to be able to select appropriate device-based physical activity instruments to accurately assess physical activity and sedentary behaviour. This review highlights the devices that could be used and the associated challenges. There are some existing guidelines for the assessment of physical activity in other populations [59,60]. There is a clear need for an agreed protocol(s) to guide the processing and analysis of data for the critical care population, therefore, to address this limitation, we have made some practical suggestions/recommendations that researchers and clinicians could use until such evidence-informed guidelines are available (Table 3). Future research may also benefit from the use of machine learning to develop protocols and assist in differentiating between different physical activity behaviours/patterns [74].

## Conclusion

The use of device-based physical activity measurement in critical illness is an active and evolving research area. A serious issue we have highlighted is that there appears to be inconsistent reporting and usage of various physical activity and sedentary behaviour parameters related to accelerometry in critical illness. This review supports the need for the development of a strategy or protocol to guide the future use of devices and standardise processing and analysis when assessing physical activity and sedentary behaviour in the population with critical illness. In addition, there is a need for researchers to establish the typical MET values required for completing common types of physical activity specifically in this population, in order to guide more effective physical activity prescription. Physical activity levels remain low in both ICU and following ICU discharge, and investment is needed to support recovery of physical activity and reduce sedentary behaviour following discharge from hospital.

## Supporting information

**S1 Text. Search strategy.**
(DOCX)

**S2 Table. Study characteristics.**
(DOCX)

**S3 Table. Correlations device measured physical activity and other health outcomes measured.**
(DOCX)

**S4 Text. PRISMA-ScR Checklist.**
(DOCX)

## Acknowledgements

The authors would like to acknowledge Kelly Coogan, assistant subject librarian, faculty of life & health science, Ulster University, in providing expertise and assisting with developing and conducting the search strategy.

## Author contributions

**Conceptualization:** Darren Murphy, Judy M Bradley, Jason J Wilson, Brenda O'Neill.

**Data curation:** Darren Murphy, Annette Henderson, Brenda O'Neill.

**Formal analysis:** Darren Murphy, Annette Henderson, Jason J Wilson, Brenda O'Neill.

**Funding acquisition:** Judy M Bradley, Jason J Wilson, Brenda O'Neill.

**Investigation:** Darren Murphy, Annette Henderson.

**Methodology:** Darren Murphy, Annette Henderson, Judy M Bradley, Bronwen Connolly, Jason J Wilson, Brenda O'Neill.

**Project administration:** Darren Murphy, Judy M Bradley, Brenda O'Neill.

**Resources:** Darren Murphy.

**Supervision:** Judy M Bradley, Jason J Wilson, Brenda O'Neill.

**Validation:** Darren Murphy, Annette Henderson, Judy M Bradley, Bronwen Connolly, Jason J Wilson, Brenda O'Neill.

**Visualization:** Darren Murphy, Annette Henderson, Judy M Bradley, Jason J Wilson, Brenda O'Neill.

**Writing – original draft:** Darren Murphy, Annette Henderson.

**Writing – review & editing:** Darren Murphy, Annette Henderson, Judy M Bradley, Bronwen Connolly, Jason J Wilson, Brenda O'Neill.

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
