## [Decision Letter · Decision Letter 0]

27 Jan 2025

PONE-D-24-55854Device-based measurement of physical activity and sedentary behaviour after critical illness: A scoping reviewPLOS ONE

Dear Dr. Murphy,

Thank you for submitting your manuscript to PLOS ONE. After careful consideration, we feel that it has merit but does not fully meet PLOS ONE’s publication criteria as it currently stands. Therefore, we invite you to submit a revised version of the manuscript that addresses the points raised during the review process.

We look forward to receiving your revised manuscript.

Kind regards,

Hidetaka Hamasaki

Academic Editor

PLOS ONE

“(DM) Funded by the Department for the Economy, Northern Ireland (DfE) PhD funding”

“The authors would like to acknowledge Kelly Coogan, assistant subject librarian, faculty of life & health science, Ulster University, in providing expertise and assisting with developing and conducting the search strategy.

Funding was provided by the Department for the Economy (DfE) Northern Ireland.”

“(DM) Funded by the Department for the Economy, Northern Ireland (DfE) PhD funding.”

Reviewers' comments:

Reviewer's Responses to Questions

**Comments to the Author**

1. Is the manuscript technically sound, and do the data support the conclusions?

Reviewer #1: Yes

Reviewer #2: Partly

Reviewer #3: Yes

Reviewer #4: Yes

2. Has the statistical analysis been performed appropriately and rigorously? 

Reviewer #1: N/A

Reviewer #2: Yes

Reviewer #3: Yes

Reviewer #4: N/A

3. Have the authors made all data underlying the findings in their manuscript fully available?

Reviewer #1: Yes

Reviewer #2: Yes

Reviewer #3: Yes

Reviewer #4: Yes

4. Is the manuscript presented in an intelligible fashion and written in standard English?

Reviewer #1: Yes

Reviewer #2: Yes

Reviewer #3: Yes

Reviewer #4: Yes

5. Review Comments to the Author

Reviewer #1: This is a precise scoping review of physical activity and sedentary behavior devices during and after critical illness and is promising for clinical application. The study methodology is transparent and reliable in that the protocol is publicly available and PRISMA-ScR compliant.

1. Practical suggestions regarding protocols

The authors emphasize the diversity of devices and data analysis methods in their discussion and conclusions. The need for standardized measurement protocols is an important point. Practical suggestions for researchers and clinicians are listed in Appendix S4. Still, if the main manuscript also mentions essential points, it will improve reader accessibility and complement the significance of this study.

2. Devices and data analysis methods specific to serious diseases

This scoping review is limited to studies during and after a serious illness. However, previous research on measuring physical activity and sedentary behavior has been conducted extensively on subjects other than those with serious illnesses. The research findings on subjects other than those with serious illness may contribute to establishing protocols for measurement devices and data analysis methods. If there are findings specific to serious illness regarding measurement devices and data analysis methods, highlighting these may enhance the novelty of the research. In addition, if there are any differences or similarities with existing protocols that target behaviors other than serious illness, these would also enhance the significance of this research.

Reviewer #2: I hope this letter finds you well. I had the opportunity to review your article titled, “Device-based measurement of physical activity and sedentary behavior after critical illness: A scoping review”, which was submitted to Plos One.

This article needs to be revised to fit the academic format.

Abstract

-. The research method written in the abstract refers to the research tool.

-. Please suggest research or analysis methods.

Introduction

-. Please change Background to introduction.

-. This reviewer does not fully recognize the necessity of this study.

-. Researchers need to logically explain ‘why’ they should conduct this research.

-. Please integrate the research objectives into one.

Method

-. Please explain ‘why’ you chose literature review to analyze the purpose of the study.

-. For example, it provides appropriate explanations of methods for literature research, such as frames, strategies, periods, and criteria.

-. The criteria for selecting prior research and data extraction were written very specifically, which is believed to increase the reliability of the study.

Results

-. The contents of Fig. 1 are not visible.

-. Please fix this by increasing the resolution so that the text can be seen.

-. Table 1, 2. Please organize and present the research materials by year.

-. The line spacing is irregular. The entire text needs to be edited.

-. I don't quite understand sentences 195-196.

-. The research results were written in detail by the researcher (level patterns, step counts, activity counts, sedentary behavior).

-. However, I question whether it has anything to do with the purpose of the study.

-. I don't understand why lines 233, 243, and 247 are spaced.

-. Overall, the research results are judged to have perfectly analyzed previous studies and to have been well organized by the researcher according to each criterion.

Discussion

-. Based on the research results, it is judged that the discussion was well conducted.

-. However, there is uncertainty as to whether the five objectives of the study were discussed.

-. The discussion needs to be organized according to the five research objectives.

-. Please modify Lines 472-475 to conform to the academic society format.

Conclusion

Please provide further explanation of the limitations of the study and suggestions for follow-up research.

-. I would like to see additional explanations regarding the academic and empirical applicability of this study.

Reviewer #3: Subject: Review of Manuscript PONE-D-24-55854

Dear Authors,

Thank you for the opportunity to review your manuscript titled "Device-based measurement of physical activity and sedentary behaviour after critical illness: A scoping review." I appreciate the effort that has gone into synthesizing the literature on this emerging research topic. The manuscript presents a well-structured scoping review and follows a rigorous methodological approach. Below, I provide my feedback regarding areas for improvement.

1. Technical Soundness and Data Support for Conclusions

The manuscript presents a technically sound scoping review on device-based measurements of physical activity and sedentary behavior in critically ill patients. The research rationale is well-explained, and the Arksey & O’Malley (2005) framework is appropriately employed for study selection. The review effectively synthesizes findings across 22 studies, covering different devices, data processing methods, and clinical outcomes.

However, a few key concerns should be addressed:

Lack of standardization in device-based data processing: The manuscript acknowledges heterogeneous data processing methods across studies but does not critically analyze the implications of such variations. A more detailed discussion on how these differences impact the comparability of findings would strengthen the manuscript.

Limited evidence on clinical impact: While the review effectively identifies device use trends, it would benefit from a more in-depth discussion on how these findings inform rehabilitation practices.

Underrepresentation of interventional studies: The manuscript notes that only one randomized control trial (RCT) was included. More discussion on the lack of high-quality trials and its implications for future research would be valuable.

2. Statistical Analysis and Data Reporting

The study does not perform a new statistical analysis but evaluates studies that have used various analytical methods.

Device comparisons lack uniform reporting: While Table 1 outlines device properties, the statistical reliability and error margins of different measurement methods should be discussed further.

The manuscript states that clinimetric properties (validity, reliability, responsiveness) were measured in only five studies but does not provide a systematic critique of these studies. A brief methodological critique of these validation studies would enhance the robustness of conclusions.

3. Data Availability and Compliance with PLOS Data Policy

The manuscript states that all relevant data are available within the manuscript and supporting materials, aligning with PLOS ONE’s open data policy. No concerns were identified regarding data accessibility.

4. Clarity and Language Quality

The manuscript is generally well-written in standard English, but minor grammatical errors and redundant phrasing could be improved. Some sentences are unnecessarily long, making them harder to follow. Examples:

Example 1 (Introduction, Lines 49-51):

"Many patients report challenges in returning to their usual activity levels."

Suggested revision: "Patients often struggle to regain pre-illness activity levels."

Example 2 (Results, Lines 312-314):

"Step counts were low, but gradually increased as patients transitioned from ICU to hospital ward to home."

Suggested revision: "Step counts were initially low but increased progressively from ICU to hospital discharge and home recovery."

Additionally, some sections lack coherence in citation formatting (see point 5).

5. Formatting and Referencing Issues

Citation inconsistencies:

The reference format appears inconsistent in certain places, e.g., citations [5] vs. [5, 6, 7]. Ensure that all references follow a consistent style.

Ensure full author names or initials are included in the reference list when required.

In-text citations:

Some references are missing the page number or DOI where necessary.

Table Formatting:

Table 1 (Device Comparison) is dense and difficult to read. Consider reorganizing it into clearer subsections (e.g., separate columns for wear time, activity parameters, and sampling frequency).

Table 2 (Measured Physical Activity and Sedentary Behavior) could benefit from bolded column headers for improved readability.

Overall Recommendation

The manuscript is technically sound, but needs minor revisions to improve clarity, reference consistency, and a more critical discussion of methodological issues in device-based measurement. Addressing these points will strengthen its contribution to rehabilitation research and critical care.

Reviewer #4: Overall Structure and Organization

The manuscript generally follows a logical structure for a scoping review, using the Arksey and O’Malley framework. The use of an inverted pyramid structure for presenting the review, with the most important information at the top, is helpful. The inclusion of a PRISMA flow diagram for study selection is also appropriate. However, there are areas within the sections that could be strengthened.

Specific Section Feedback

• Abstract: The abstract effectively summarizes the background, aims, methods, results, and conclusions of the review. However, the conclusion could be strengthened by explicitly stating the need for standardizing data processing and analysis protocols.

• Introduction/Background: This section introduces the problem of reduced physical activity after critical illness and the potential benefits of physical activity for rehabilitation. The background effectively establishes the importance of measuring physical activity in this population. However, the introduction could benefit from an earlier and clearer statement of the need for device-based measurement and the challenges associated with it.

• Aims and Objectives: The aims and objectives are clearly stated and well-defined. The questions outlined are relevant and directly addressed by the review.

• Methods: The methods section is comprehensive and outlines the search strategy, eligibility criteria, study selection process, and data extraction methods. The use of four electronic databases (Medline, ProQuest, Scopus, and CINAHL) is appropriate. The process of duplicate removal and study selection is clearly described. The use of Covidence as a screening tool is also noted. However, it is mentioned that the protocol was published in advance, but a statement indicating if there were deviations from the protocol is missing.

• Results: The results section clearly presents the study characteristics, device-based instruments used, parameters for physical activity, clinimetric properties, and associations with health-related outcomes. The use of tables and a flow diagram is very effective in presenting the data. However, there is some inconsistent formatting in the tables, as previously mentioned [see prior response and 142, 144, 147], where text appears to run into each other.

o Study Characteristics: This section provides a good overview of the included studies, but it could further emphasize the heterogeneity of the populations and study designs, which can limit generalizability.

o Device-based Instruments: The section provides a good overview of the devices used. It highlights that the Actiwatch Spectrum, ActivPAL3, and SenseWear armband were the most commonly used. The manuscript also highlights the inconsistent device placement (dominant/non-dominant limb). This variation should be highlighted as it can affect the results.

o Parameters of Physical Activity: This section identifies the variation in how physical activity and sedentary behavior were measured and reported (step counts, activity counts, time in activity, intensity of activity). This inconsistency is a major issue that needs to be addressed in the discussion.

o Clinimetric Properties: The lack of information regarding the reliability, responsiveness, and MCIDs is correctly identified. This lack of information needs to be addressed in the discussion as a major issue.

o Association with Health Parameters: The associations between physical activity and other health-related parameters are detailed, but some of the findings appear inconsistent across studies. The authors could potentially discuss why some studies show an association while others do not.

o Levels and Patterns of Physical Activity: This section effectively presents the low levels of physical activity and high levels of sedentary behavior in the ICU and post-discharge. The specific data included in Table 2 is useful to highlight the low activity levels throughout the trajectory of care.

• Discussion: The discussion section is thorough, summarizing key findings and highlighting the need for standardized protocols. The manuscript correctly highlights the inconsistencies in defining physical activity, sedentary behavior, intensity of activity, and thresholds for data analysis. The discussion also mentions the use of various sampling frequencies and the impact of epoch length. The authors could expand on the practical implications of these inconsistencies for clinicians and researchers. The importance of validating MET values for the critical care population is also emphasized. The discussion also highlights the potential impact of the use of low frequency filters on step counts. The discussion also mentions the need for future research and the need for an agreed protocol. The strengths and limitations of the review are not explicitly discussed in this section.

• Conclusion: The conclusion effectively summarizes the key findings of the review, reiterating the need for a standardized approach to device-based physical activity measurement in critical illness. However, the conclusion could further emphasize the need to validate MET values.

• Tables and Figures:

o Table 1: This table provides a comprehensive overview of the included studies, including devices used, wear time, and activity parameters. However, some of the data appears to be running into the adjacent cells of the table.

o Table 2: This table provides a summary of the measured physical activity and sedentary behavior. The information presented is thorough, however, there is an issue with the formatting.

o Figure 1: The PRISMA flow diagram is clear and accurately presents the study selection process.

Potential Errors to Address

• Inconsistent Reporting: The manuscript highlights the inconsistent reporting of various parameters related to the device and data processing, but this is not reflected in a limitation of the study, nor is it highlighted as a major issue.

Areas for Improvement

• Standardization: The manuscript should provide more concrete recommendations for standardization, such as suggesting specific epoch lengths, sampling frequencies, and MET thresholds for the critical care population.

• Clinimetric Properties: The authors should further emphasize the need for more research focused on validating device performance for critical care patients. The importance of establishing reliability, responsiveness, and MCIDs for physical activity and sedentary behavior in this population should be highlighted.

• MET Values: The manuscript should emphasize the need for the validation of appropriate MET values for this population, as previously discussed.

• Strengths and Limitations: The manuscript should explicitly state the strengths and limitations of the review process in the discussion section. This should include a discussion of the study designs of the included studies.

• Practical Implications: The authors could expand on the practical implications of their findings for both researchers and clinicians.

• Formatting: The formatting issues in the tables need to be addressed to ensure that the information is clearly displayed.

• Study Protocol Deviations: The authors should include a statement in the methods indicating if there were deviations from the published protocol.

Conclusion

The manuscript is a well-conducted scoping review that highlights the important issue of measuring physical activity in critically ill patients. By addressing the highlighted issues, particularly related to the standardization of data processing, establishing clinimetric properties, validating MET values for this population, and resolving formatting issues, the manuscript will be suitable for publication. The key areas to address would be the standardization of data processing and analysis, validation of MET values, and establishment of clinimetric properties for the critical care population.

6. PLOS authors have the option to publish the peer review history of their article (what does this mean? ). If published, this will include your full peer review and any attached files.

**Do you want your identity to be public for this peer review?** For information about this choice, including consent withdrawal, please see our Privacy Policy .

Reviewer #1: No

Reviewer #2: No

Reviewer #3: No

Reviewer #4: **Yes: ** Abdulmalik Alilu Abubakar

---

## [Author Response · Author response to Decision Letter 1]

12 Mar 2025

Reviewer #1:

1. Practical suggestions regarding protocols

The authors emphasize the diversity of devices and data analysis methods in their discussion and conclusions. The need for standardized measurement protocols is an important point. Practical suggestions for researchers and clinicians are listed in Appendix S4. Still, if the main manuscript also mentions essential points, it will improve reader accessibility and complement the significance of this study.

(R1.1) Response: We agree that the main manuscript should mention important points (e.g. the need for standardised measurement protocols). We have checked to ensure essential points are included in the Discussion section. See line 436 “In addition, the location the device is worn (e.g. thigh, wrist) alongside the dominant/non-dominant side worn have been shown to impact on important PA parameters, such as step counts. For example, daily steps have been shown to be significantly difference between wrist and hip locations in adults wearing a pedometer [54], while another study [55] demonstrated dominant wrist placement resulted in 1,253 more daily steps compared to the non-dominant side (p=0.006)”

In addition, we have further signposted to Table 3 (previously appendix 1; appendix 1 has now been relabelled as Table 3 and moved to the main body of the manuscript to improve accessibility).

2. Devices and data analysis methods specific to serious diseases

This scoping review is limited to studies during and after a serious illness. However, previous research on measuring physical activity and sedentary behaviour has been conducted extensively on subjects other than those with serious illnesses. The research findings on subjects other than those with serious illness may contribute to establishing protocols for measurement devices and data analysis methods. If there are findings specific to serious illness regarding measurement devices and data analysis methods, highlighting these may enhance the novelty of the research. In addition, if there are any differences or similarities with existing protocols that target behaviours other than serious illness, these would also enhance the significance of this research.

(R1.2) Response: The reviewer is correct, previous research on measuring physical activity and sedentary behaviour has been conducted extensively on subjects other than those with serious illnesses. In contrast, device-based measurement of PA is relatively new to the area of critical illness recovery with more than 50% (n=12) of the papers included in this scoping review published in the last five years. A recent paper on “wearable devices” also highlights and concurred that there are challenges related to device processing that need to be overcome in order to guide decision making and clinical practice for the critical care population (Angelucci et al 2025). Therefore, where possible, we have embedded the learning from subjects other than those with serious illnesses in our manuscript. For example, only one included study used a less typical approach to normal MET values as the threshold for physical activity 1 MET as opposed to 1.5 MET. We have contrasted this with other non-ICU / serious illness populations to highlight the use of MET values. We then discussed this in the context of other populations who used a similar approach (see lines 417-426)

Furthermore, Table 3 is informed by the papers reviewed within the scoping review and when these do not offer data to guide device-based physical activity measurement and analysis, then references in subjects other than those with serious illnesses have been used (e.g. cardiovascular patients and older adults).

Reviewer #2:

Abstract

The research method written in the abstract refers to the research tool.

Please suggest research or analysis methods.

(R 2.1) Response: We have added in an extra line to detail the research methods.

Please see line (35-37) of the Abstract methods “Screening and data extraction was conducted by two independent reviewers. Data were analysed descriptively by summarising and describing results that linked to the review questions.”

Introduction

Please change Background to introduction.

This reviewer does not fully recognize the necessity of this study.

Researchers need to logically explain ‘why’ they should conduct this research.

Please integrate the research objectives into one.

(R 2.2) Response: We have added text to the introduction to further emphasise and clarify the rationale for the review. (Line 68-72) “The consequences of critical illness, as well as sedentary behaviour, have the potential to impact patients' recovery and return to activity after hospital discharge. However, physical activity is not routinely measured in this population and it is not clear which device should be used. There is also limited data to quantify and describe levels of physical activity and sedentary behaviour after critical illness.”

Further explanation is included in lines 74-76 “There is increasing awareness about the need to evaluate and promote rehabilitation and physical activity in the critical care population during and after hospitalisation [11]”; as well as lines 87-89 “This review will inform the potential use and implementation of device-based measurement of physical activity and sedentary behaviour following critical illness in future research and in clinical practice.”

For the scoping review questions listed under the “Aims”, these have now been incorporated into the Methods section, in line with scoping review methodologies recommendations. We have also now included abbreviated text in the “Aims” section (please see below and Lines 95-100).

“The objectives were, in patients following critical illness, to identify device-based instruments and the parameters used to describe patterns and levels of physical activity and sedentary behaviour, report device clinimetric properties; describe the relationship between physical activity and sedentary behaviour with other health-related outcomes, and explore the levels and patterns of physical activity and sedentary behaviour across the trajectory of recovery.”

Method

Please explain ‘why’ you chose literature review to analyze the purpose of the study.

For example, it provides appropriate explanations of methods for literature research, such as frames, strategies, periods, and criteria.

The criteria for selecting prior research and data extraction were written very specifically, which is believed to increase the reliability of the study.

(R 2.3) Response: We are happy to clarify the rationale for using a scoping review methodology. The Methods section has been updated with additional text to further clarify the rationale for choosing scoping review methodology (Line 103-105) “We chose a scoping review as the most appropriate review method in order to synthesise the evidence to address specific questions relating to device-based measurement of physical activity and sedentary behaviour [15]”.

In addition, we adhered to the PRISMA-ScR criteria to robustly report the results.

Results

The contents of Fig. 1 are not visible.

Please fix this by increasing the resolution so that the text can be seen.

-Table 1, 2. Please organize and present the research materials by year.

(R 2.4) Response: Thanks for spotting this. Figure 1 has now been updated to ensure that it complies with the PLOS One formatting and resolution requirements. This should improve the clarity of the text. Tables 1 & 2 have both been completed/amended to accommodate your suggestion of presenting the research materials by year.

The line spacing is irregular. The entire text needs to be edited.

(R 2.5) Response: We apologise for this oversight; this has now been updated to keep spacing consistent throughout the Results section.

I don't quite understand sentences 195-196.

(R 2.6) Response: We have now clarified the smallest sample (n=8) and the largest sample size (n=715). See line 213-215 “The population sample sizes in the included studies ranged between 8 [24] and 715 [23] participants.”

-. The research results were written in detail by the researcher (level patterns, step counts, activity counts, sedentary behavior).

-. However, I question whether it has anything to do with the purpose of the study.

(R 2.7) Response: Thank you for this valid query. The study team carefully thought through our research questions before submitting our protocol, as well as considering the proposed value that this review could have for future researchers and clinical practice. While discussing the objectives and research questions of our scoping review, we felt that to fully address the study aims, we needed to firstly describe what devices and processing rules have been utilised in those with critical illness; and the team agreed that it would add important context for the reader to include knowledge about the typical levels of physical activity and sedentary behaviour reported when using these particular devices and parameters across the patient journey (i.e. from admission to many months post-discharge).

-. I don't understand why lines 233, 243, and 247 are spaced.

(R 2.8) Response: Thank you for highlighting this. This has now been adjusted to create appropriate paragraphs (see 247-256).

-. Overall, the research results are judged to have perfectly analyzed previous studies and to have been well organized by the researcher according to each criterion.

(R 2.9) Response: The authors appreciate this kind feedback.

Discussion

-. Based on the research results, it is judged that the discussion was well conducted. However, there is uncertainty as to whether the five objectives of the study were discussed. The discussion needs to be organized according to the five research objectives

(R2.10) Response: We are happy to amend the Discussion section to ensure all objectives are clearly discussed. To further help organise these within the Discussion section, we have now added subtitles to profile the discussion of each of the five objectives e.g. Lines 382, 411.

-. Please modify Lines 472-475 to conform to the academic society format.

(R 2.11) Response: Well-spotted, we have now made suitable adjustments for these lines

Conclusion

-.Please provide further explanation of the limitations of the study and suggestions for follow-up research.

(R 2.12) Response: Thank you, we may have under-reported these and so further detail has been added to the strengths and limitations section. See lines 534-541. “The strengths of this scoping review include a comprehensive search strategy and pre-identified rigorous methodology, as well as the inclusion of studies across the entire recovery trajectory following ICU. The review protocol was registered a priori to ensure transparency. Limitations of this scoping review were that we did not contact authors to ask whether further details about their data processing could be made available and our results were limited by the lack of a standardised approach to measurement making it difficult to compare across studies.”

The second part of your query about follow-up research has been addressed in lines 547-551 in the Discussion section and also Lines 561-566 with the Conclusion.

-. I would like to see additional explanations regarding the academic and empirical applicability of this study.

(R 2.13) Response: This is certainly important, and we feel this has now been addressed by adding additional rationale in the Introduction section, additional text in the Discussion section, and directing the reader to Table 3 (previously Appendix 1), along with the explanation and recommendations in the Conclusion section (558-566). The manuscript is already quite lengthy, particularly after addressing your comments along with the other reviewers, but if you have a thought on a further area we have not covered, then we certainly welcome additional feedback from the reviewer.

Reviewer #3:

Lack of standardization in device-based data processing: The manuscript acknowledges heterogeneous data processing methods across studies but does not critically analyze the implications of such variations. A more detailed discussion on how these differences impact the comparability of findings would strengthen the manuscript.

(R 3.1) Response: This is certainly an important issue to highlight within our manuscript. We feel this has now been more fully addressed in lines 430-436 and 443-448 for processing decisions related to epoch length and sampling frequency.

Limited evidence on clinical impact: While the review effectively identifies device use trends, it would benefit from a more in-depth discussion on how these findings inform rehabilitation practices.

(R 3.2) Response: Again, this is an important aspect to consider to increase the utility of this particular article for practice. We feel this has been now addressed within the Discussion section (e.g. lines 547-554. This has also been addressed within Table 3, by providing practical considerations and suggestions for clinicians to utilise in practice. Also, the Discussion section on “Levels of physical activity and sedentary behaviour” now has additional commentary on how the findings can be applied (see lines 528-532).

Underrepresentation of interventional studies: The manuscript notes that only one randomized control trial (RCT) was included. More discussion on the lack of high-quality trials and its implications for future research would be valuable.

(R 3.3) Response: We agree that the lack of high-quality RCTs is worth noting. We have further discussed this and also the wider implications (see responses 3.1 and 3.2 above and also the additional text in lines 525-532: “High-quality experimental studies are required to establish cause and effect relationships between physical activity and sedentary behaviour with important patient outcomes. At present, nearly all research is observational in nature. Therefore, there is a clear need for future experimental studies exploring the impact of changing physical activity and sedentary behaviour on important patient outcomes. Investigation is also needed to test the best way to support recovery of physical activity and reduction of excessive sedentary behaviour following discharge from hospital.”

Statistical Analysis and Data Reporting

Device comparisons lack uniform reporting: While Table 1 outlines device properties, the statistical reliability and error margins of different measurement methods should be discussed further.

(R3.4) Response: We agree that this is certainly something that must be clearly highlighted. Therefore, we have now added in a paragraph early in the Discussion section to talk about needing to consider measurement error when using different devices (Discussion, Lines 396-409). “While various accelerometers have been utilised in individuals admitted to ICU, it is important for researchers and clinicians to be aware that there is likely to be a certain amount of measurement error with each device. For example, Baldwin and colleagues [24] compared activPAL-measured outcome variables with direct observation in ICU survivors. The authors found that although sit-to-stand transitions, sedentary behaviour and upright time were well-recognised by the activPAL, there were under-estimations of standing and stepping time compared to direct observation (22% and 19%, respectively). In healthy populations, free-living step counts have been shown to vary across seven activity monitors by around 1700 steps/day (ranging from 1265 to 2275 steps/day) [42]. This shows that even though two activity monitors are supposedly measuring the same variable, there are likely to be error margins to consider. Unfortunately, few studies have currently explored validity and reliability of different devices in ICU survivors.”

Clarity and Language Quality

The manuscript is generally well-written in standard English, but minor grammatical errors and redundant phrasing could be improved.

Some sentences are unnecessarily long, making them harder to follow. Examples:

Example 1 (Introduction, Lines 49-51):

"Many patients report challenges in returning to their usual activity levels."

Suggested revision: "Patients often struggle to regain pre-illness activity

---

## [Decision Letter · Decision Letter 1]

21 Mar 2025

Device-based measurement of physical activity and sedentary behaviour after critical illness: A scoping review

PONE-D-24-55854R1

Dear Dr. Murphy,

We’re pleased to inform you that your manuscript has been judged scientifically suitable for publication and will be formally accepted for publication once it meets all outstanding technical requirements.

Kind regards,

Hidetaka Hamasaki

Academic Editor

PLOS ONE

Additional Editor Comments (optional):

Reviewers' comments:

Reviewer's Responses to Questions

**Comments to the Author**

1. If the authors have adequately addressed your comments raised in a previous round of review and you feel that this manuscript is now acceptable for publication, you may indicate that here to bypass the “Comments to the Author” section, enter your conflict of interest statement in the “Confidential to Editor” section, and submit your "Accept" recommendation.

Reviewer #1: All comments have been addressed

Reviewer #2: (No Response)

2. Is the manuscript technically sound, and do the data support the conclusions?

Reviewer #1: Yes

Reviewer #2: (No Response)

3. Has the statistical analysis been performed appropriately and rigorously? 

Reviewer #1: N/A

Reviewer #2: (No Response)

4. Have the authors made all data underlying the findings in their manuscript fully available?

Reviewer #1: Yes

Reviewer #2: (No Response)

5. Is the manuscript presented in an intelligible fashion and written in standard English?

Reviewer #1: Yes

Reviewer #2: (No Response)

6. Review Comments to the Author

Reviewer #1: The authors have made sufficient revisions in response to my comments. Thank you for your research, which is worthy of publication.

Reviewer #2: -. The content of this study was reviewed by four reviewers.

-. This reviewer believes that the researcher is fully aware of the reviewer's content.

-. It is also judged that the reviewer's content has been sufficiently revised.

-. In particular, it is judged that the introduction, methods, and discussion have been appropriately revised in response to the reviewer's revisions.

-. However, there are still parts of the paper format that are not compatible with the academic society format.

-. Please revise the paper format before final publication.

7. PLOS authors have the option to publish the peer review history of their article (what does this mean? ). If published, this will include your full peer review and any attached files.

**Do you want your identity to be public for this peer review?** For information about this choice, including consent withdrawal, please see our Privacy Policy .

Reviewer #1: No

Reviewer #2: No

---

## [Editor Report · Acceptance letter]

PONE-D-24-55854R1

PLOS ONE

Dear Dr. Murphy,

I'm pleased to inform you that your manuscript has been deemed suitable for publication in PLOS ONE. Congratulations! Your manuscript is now being handed over to our production team.

Kind regards,

on behalf of

Dr. Hidetaka Hamasaki

Academic Editor

PLOS ONE
